# A Study of Phytochemistry, Genoprotective Activity, and Antitumor Effects of Extracts of the Selected Lamiaceae Species

**DOI:** 10.3390/plants10112306

**Published:** 2021-10-27

**Authors:** Mariana Oalđe Pavlović, Stoimir Kolarević, Jelena Đorđević, Jovana Jovanović Marić, Tanja Lunić, Marija Mandić, Margareta Kračun Kolarević, Jelena Živković, Ana Alimpić Aradski, Petar D. Marin, Katarina Šavikin, Branka Vuković-Gačić, Biljana Božić Nedeljković, Sonja Duletić-Laušević

**Affiliations:** 1Department of Plant Morphology and Systematics, Faculty of Biology, Institute of Botany and Botanical Garden “Jevremovac”, University of Belgrade, Studentski trg 16, 11070 Belgrade, Serbia; alimpic.ana@bio.bg.ac.rs (A.A.A.); pdmarin@bio.bg.ac.rs (P.D.M.); sduletic@bio.bg.ac.rs (S.D.-L.); 2Centre for Genotoxicology and Ecogenotoxicology, Department of Microbiology, Faculty of Biology, Institute of Botany and Botanical Garden “Jevremovac”, University of Belgrade, Studentski trg 16, 11070 Belgrade, Serbia; stoimir.kolarevic@ibiss.bg.ac.rs (S.K.); b3013_2017@stud.bio.bg.ac.rs (J.Đ.); b3010_2016@stud.bio.bg.ac.rs (J.J.M.); brankavg@bio.bg.ac.rs (B.V.-G.); 3Department of Hydroecology and Water Protection, Institute for Biological Research “Siniša Stanković”, National Institute of Republic of Serbia, University of Belgrade, Bulevar Despota Stefana 142, 11070 Belgrade, Serbia; margareta.kracun@ibiss.bg.ac.rs; 4Institute for Multidisciplinary Research, University of Belgrade, Kneza Višeslava 1, 11070 Belgrade, Serbia; 5Faculty of Biology, Institute of Physiology and Biochemistry “Ivan Djaja”, University of Belgrade, Studentski trg 16, 11070 Belgrade, Serbia; b3018_2019@stud.bio.bg.ac.rs (T.L.); b3022_2019@stud.bio.bg.ac.rs (M.M.); biljana@bio.bg.ac.rs (B.B.N.); 6Institute for Medicinal Plants Research “Dr. Josif Pančić”, Tadeuša Košćuška 1,11070 Belgrade, Serbia; jzivkovic@mocbilja.rs (J.Ž.); ksavikin@mocbilja.rs (K.Š.)

**Keywords:** Lamiaceae, phenolic compounds, antioxidant activity, antigenotoxicity, genoprotective effect, antitumor activity

## Abstract

This study was designed to evaluate the genoprotective, antigenotoxic, as well as antitumor potential of methanolic, ethanolic, and aqueous extracts of *Melissa officinalis*, *Mentha* × *piperita*, *Ocimum basilicum*, *Rosmarinus officinalis*, *Salvia officinalis*, and *Satureja montana* (Lamiaceae), in different model systems. The polyphenols in these extracts were quantified both spectrophotometrically and using HPLC-DAD technique, while DPPH assay was used to assess the antioxidant activity. The genoprotective potential was tested on pUC19 *Escherichia coli* XL1-blue, and the antigenotoxicity on *Salmonella typhimurium* TA1535/pSK1002 and human lung fibroblasts, while the antitumor activity was assessed on colorectal cancer cells. Rosmarinic acid, quercetin, rutin, and luteolin-7-*O*-glucoside were among the identified compounds. Methanolic extracts had the best DPPH-scavenging and SOS-inducing activities, while ethanolic extracts exhibited the highest antigenotoxicity. Additionally, all extracts exhibited genoprotective potential on plasmid DNA. The antitumor effect was mediated by modulation of reactive oxygen species (ROS), nitric oxide (NO) production, and exhibition of genotoxic effects on tumor cells, especially with *O. basilicum* ethanolic extract. Generally, the investigated extracts were able to provide antioxidant protection for the acellular, prokaryotic, and normal human DNA, while also modulating the production of ROS and NO in tumor cells, leading to genotoxicity toward these cells and their decrease in proliferation.

## 1. Introduction

In living cells, free radicals, such as reactive oxygen (ROS) and nitrogen species (RNS), are continuously produced as a follow-up of both metabolism and environment. Since they also function as signaling molecules, ROS and RNS production is genetically programmed in order to protect the body and alleviate inflammatory processes. However, if their overproduction surpasses the antioxidant capacity of the cell, these radicals begin to oxidize protein and lipid cell elements, and inflict physical and chemical damage to the DNA molecule [1], resulting in modified bases, single and double-strand breaks, and DNA-DNA and DNA-protein crosslinks [2]. Moreover, ROS affects the DNA-binding ability of a number of transcription factors, depriving the cell of the efficient repair system [1]. Ultimately, these events lead to a disturbance in the cell physiology, various mutations, and excessive tissue damage. Hence, in order for the normal cells to survive these internal, as well as many external attacks, a vast number of biological processes must be initiated at the cellular, tissue, organ, and organism levels [2]; otherwise, free radicals cause oxidative stress, which is directly connected to the development and progression of a plethora of diseases, cancer being one of them [3,4,5]. Therefore, timely application of antioxidant substances is essential in order to protect and preserve the integrity of the DNA and other cell components, to prevent and/or control tumorigenesis, and to inhibit the growth of a tumor, as well as to improve the general well-being of the organism [1]. Multiple drugs are being used in tumor therapy; however, tumor cells are prone to various gene mutations, which can influence the efficiency of the administered synthetic drugs [6]. Hence, there is a growing interest in nutraceuticals with the potential to restore the imbalance caused by oxidative stress, to regenerate the non-tumor tissue and also induce cancer cell death [5,7,8]. In spite of the presence of an alarming number of diseases in our population today, plants represent an invaluable source of effective agents with medicinal potential [4], already being thoroughly investigated and applied in therapy.

Lamiaceae is one of the most numerous families of flowering plants, comprising 236 genera and about 7200 species [9]. A number of these plants are being used in traditional medicine and cookery since ancient times. Their healing properties are reflected in the abundance of biologically active compounds found in both their essential oils and extracts [9,10,11,12,13,14]. Wink [15] pointed out that most of the Lamiaceae species listed as antioxidants belong to the subfamily Nepetoideae, such as basil, lemon balm, peppermint, oregano, rosemary, sage, etc. Modern pharmacological studies demonstrate that *Melissa officinalis* (*Mo*, lemon balm) exhibit several biological activities, including antioxidant, hypoglycemic, antimicrobial, antitumor, antidepressant, anxiolytic, anti-nociceptive, anti-inflammatory, and spasmolytic properties [16,17]. *Mentha* × *piperita* (*Mp*, peppermint) is reported to have antimicrobial, antiviral, insecticidal, antioxidant, antiamoebic, antihemolytic, antiallergenic, and antitumor potential [9], and it is used as an adjuvant in gastrointestinal infections, digestive and respiratory disorders, nausea, nervous or mental fatigue, and mouth hygiene [18]. *Ocimum basilicum* (*Ob*, basil) is commonly used for its anxiolytic, sedative, anti-colitis effects, antioxidant, antibacterial, and cytotoxic activity, as well as antihepatotoxic, hypoglycemic, antihypertensive, vasorelaxant, neuroprotective, and anti-inflammatory effects [19,20]. *Rosmarinus officinalis* (*Ro*, rosemary, syn. *Salvia rosmarinus* Spenn.) is known to have antibacterial, antiviral, antioxidant, anti-inflammatory, antispasmodic, diuretic, and chemopreventive properties, and is used to treat minor wounds, rashes, headache, dyspepsia, and circulation problems [9,21]. Studies on *Salvia officinalis* (*So*, sage) have revealed a wide range of pharmacological activities, including antitumor, anti-inflammatory, anti-nociceptive, antioxidant, antimicrobial, antineurodegenerative, hypoglycemic, hypolipidemic, and antimutagenic effects [12,22]. *Satureja montana* (*Sm*, winter savory) is reported to possess antiseptic, aromatic, carminative, digestive, and expectorant properties, and it is used in the treatment of insect bites [9].

This study is part of a long-term and comprehensive research involving various biological activities of extracts of plants from the Lamiaceae family. The main objective of this study was to assess the ability of methanolic, ethanolic, and aqueous extracts of six Lamiaceae species in preserving the DNA integrity of healthy model systems against oxidative damage, while also testing the antitumor potential of the ethanolic extracts on colorectal cancer cell line.

According to the literature data, polyphenols were proposed as the main antioxidant compounds of extracts from the Lamiaceae species [5,20]; hence, it has been hypothesized that, among others, *Mo*, *Mp*, *Ob*, *Ro*, *So*, and *Sm* will also have significant antioxidant and genoprotective activity toward healthy cells and genotoxic effect toward tumor cells due to high amounts of these secondary metabolites. For the purpose of this study, phytochemical analysis of different polyphenols, antioxidant, antigenotoxic, genoprotective, cytotoxic, and genotoxic, as well as ROS-and NO (Nitric Oxide)-modulating, activities were determined. Moreover, to the best of our knowledge, this is the first comprehensive report regarding the genoprotective and antigenotoxic activities of extracts of these Lamiaceae species in acellular and prokaryotic models (pUC19 *E. coli* XL1-Blue and *Salmonella typhimurium* TA1535/pSK1002), as well as the genotoxic activity of these species on colorectal cancer cells.

## 2. Results and Discussion

### 2.1. Phytochemical Characterization

Hot water and ethanol were chosen as solvents due to the fact that medicinal plants are widely consumed as teas and tinctures. Moreover, methanol was chosen as a commonly used solvent with proven efficiency for the extraction of compounds with powerful antioxidant capacity.

The contents of total phenolic compounds, phenolic acids, flavonoids, and flavonols were expressed as standards’ equivalents and the results are presented in Table 1 and Appendix A. For each test, the extracts were analyzed in three concentrations (100, 200, and 500 µg/mL), in order for the obtained results to be comparable with the results from the DPPH test. Moreover, this was a mean of validation of the applied methodology, since there was a concentration gradient observed for the investigated extracts. However, since the DPPH activity of the extracts will only be analyzed at 250 µg/mL, the content of polyphenolic compounds will also be presented at this concentration (Table 1).

Generally, methanolic extracts were the richest in polyphenolic compounds according to the PAC, TFC, and FC analysis. Overall, at the highest applied concentration, ethanolic extracts were the ones with the highest TPC of all the examined samples, followed by the aqueous ones, while the methanolic extracts had the lowest TPC values. However, exceptions were both methanolic and aqueous extracts of *Mo* and *Mp*, which were found to have the highest TPC (Appendix A). Moreover, statistical analysis showed that the extraction solvent had an influence on the level of quantified TPC. Actually, the analysis showed that the aqueous extract of *Ob*, *Ro*, and *Sm* had significantly more phenolic compounds than the alcoholic extracts of these species, which was contrary to the results found for *So* extracts. Mabrouki et al. [23] found lower TPC values for ethanolic extract of *Mo* from Portugal compared to our results, which may be attributed to different ecological factors at *Mo* growth sites, as well as to the applied extraction protocol. Stagos et al. [24] found that *Mentha* sp. aqueous extracts showed higher TPC in comparison with the methanolic ones, while, for *So* extracts, it was the other way around, which is in compliance with the results presented in this study. Sytar et al. [25] reported that *So* methanolic extract had higher TPC than *Ro*, and *Ro* had higher TPC than *Mo*, which is different than the results obtained in this study (*Mo* > *So* > *Ro*).

For PAC, in general, the obtained values decreased in the following order: methanolic > ethanolic > aqueous. Statistical analysis revealed that alcoholic extracts had significantly higher PAC than the aqueous ones, except in the case of *So* (Table 1). There is not enough available literature data on spectrophotometrically determined PAC; however, this phytochemical content is usually quantified using HPLC method. Skendi et al. [26] found that methanolic extracts (70% methanol) have higher PAC in comparison with aqueous extracts of selected Greek Lamiaceae representatives, which is in line with the results presented in this paper. When analysing methanolic extracts, they found that *Mo* has a higher PAC than *Sm*, followed by *Ro*. On the other hand, that order when using aqueous extracts was: *Mo* > *Ro* > *Sm*. These findings are in compliance with the ones presented in the Appendix A and Table 1, despite the fact that different methodological approaches were used.

In general, methanolic extracts showed the highest TFC and FC of all the examined samples, with the exception of the aqueous extract of *Sm* that had significantly higher TFC and FC than its alcoholic extracts (Table 1). Specifically, both methanolic and ethanolic extracts of *Mp* and *So* had the highest TFC and FC in comparison with the other samples (Table 1 and Appendix A). Sytar et al. [25] found similar results to the ones presented in our study and estimated that *So* have higher TFC than *Ro*, and *Ro* have higher TFC than *Mo*. Skendi et al. [26], however, found the following order of TFC in the methanolic extracts: *Sm* > *Mo* > *Ro*, which differs from the order in Table 1 (*Ro* > *Sm* > *Mo*). Their TFC decreased in the aqueous extracts as following: *Mo* > *Sm* > *Ro*, while in this study *Ro* had the highest TFC, while *Mo* had the lowest TFC. Apart from Lee et al. [27], who reported that acetone extract of *Ob* had higher FC than *Ro* and *So*, there is no available literature data on FC assay for the six investigated Lamiaceae species.

A more detailed chemical analysis discovered that the tested extracts have a rather high content of rosmarinic acid (up to 86.76 mg/g dry extracts for the methanolic extract of *Mo*) and also a rather high content of flavonoids, such as quercetin (up to 12.43 mg/g dry extracts for the methanolic extract of *Mo*), rutin (up to 10.84 mg/g dry extracts for the aqueous extract of *Ro*), naringin (up to 12.83 mg/g dry extracts for the methanolic extract of *So*), and luteolin-7-*O*-glucoside (up to 22.82 mg/g dry extracts for the methanolic extract of *So*). Statistical analysis revealed that the applied extraction solvent significantly influences the quantity and quality of the identified phenolic compounds (Table 2), which has also been proven in earlier studies [28,29]. These results are in accordance with the existing literature data, which suggest that these particular phenolic compounds are quite common in the extracts of interest [30,31,32].

Finally, TPC had a strong correlation with PAC, TFC, and FC (r = 0.90, 0.78, and 0.69, respectively). PAC showed a moderate correlation with TFC and FC, r = 0.65 and 0.56, respectively, while, as expected, TFC correlated very well with FC (r = 0.93). On the other hand, the results of the quantified phenolic compounds had a low to moderate correlation with the total content of phenolics determined spectrophotometrically, while a strong correlation was found only for rosmarinic acid and TPC (r = 0.68) (Table 3).

### 2.2. Antioxidant Activity In Vitro

The biological activity of these plants has previously been thoroughly investigated mostly for their essential oils; hence, our literature survey showed that there is insufficient comprehensive information on the bioactivities of their extracts.

The in vitro antioxidant activity of the extracts at 100, 250, and 500 µg/mL was determined by DPPH assay (Table 4). The results showed a concentration dependency; however, it was noticed that, in the majority of cases, the reaction mixture reached a plateau at the highest applied concentration, while the results were best distinguished at the concentration of 250 µg/mL; thus, the latter concentration will be discussed hereinafter. Statistical analysis showed that the extraction solvent might have an impact on the displayed DPPH-scavenging activity. For instance, at 250 µg/mL, in the case of *Mo* and *Sm*, there was no significant difference found between the activity of methanolic, ethanolic, and aqueous extracts, while, in the case of *Mp*, the alcoholic extracts were significantly more active than the aqueous one. Overall, methanolic extracts showed the highest effectiveness against the DPPH radical, while aqueous extracts were less effective in comparison with the alcoholic ones. Of all the investigated extracts, the methanolic one of *Mp* showed the highest activity (93.40%), whereas the aqueous one of *Ob* had the lowest activity against DPPH radical (27.68%). Moreover, the results showed that most of the tested samples at the concentration of 250 µg/mL had significantly higher DPPH-scavenging activity than the positive controls BHA and BHT (except *Ob* extracts, *Ro* ethanolic, and *So* aqueous extract).

Sytar et al. [25] have found that *Mo* methanolic extract inhibited less DPPH radicals in comparison to *Ro*, whilst *So* extract expressed the highest percentage of inhibition. Their findings were not completely in accordance with our results, where *Mo* exhibited higher activity compared to *Ro* and *So*. Skendi et al. [26], on the other hand, found that *Sm* methanolic extracts had higher activity than *Ro*. Vladimir-Knežević et al. [10] reported the following (decreasing) order of ethanolic extracts activity: *So* > *Ro* > *Mo* > *Mp* > *Sm*, which is not entirely in accordance with our results. Elansary et al. [33], on the other hand, found similar results to the ones presented in this study, highlighting the DPPH-scavenging activity of *Ro* over *Ob*.

It is a well-known fact that the chemical composition of plant extracts is closely related to their biological activities; hence, it was once again proven that the results obtained in this assay correlated very well with TPC, PAC, and TFC of the investigated extracts. The correlation coefficient ranged from 0.75 to 0.89, and, according to Taylor [34], this represents a strong correlation. A moderate correlation was found between DPPH assay and FC (r = 0.65) (Table 3). This correlation analysis confirmed that phenolics in general contribute to the antioxidant activity of the examined plants, displayed in the DPPH assay. Moreover, the correlation between the quantified phenolics and the DPPH assay was found to be low to moderate, and, from the Table 3, it can be concluded that caffeic acid and its ester rosmarinic acid are mostly responsible for the displayed DPPH-scavenging activity of the tested extracts. The latter is not surprising giving that these phenolic compounds are known to exhibit numerous and powerful bioactivities [35,36]. However, it is possible that a more detailed chemical analysis could reveal a larger number of compounds with the ability to boost the DPPH-scavenging potential of these extracts.

### 2.3. Genoprotective Activity in Acellular System

Another way to determine the antioxidant activity was to test the genoprotective effect of the extracts on plasmid DNA. Hydroxyl radical (OH^-^), generated from UV photolysis of hydrogen peroxide, leads to cutting of DNA strand, which results in splitting of the supercoiled circular DNA into open circular/linear form [37]. It has been reported that the determination of the oxidative cleavage of a double-stranded DNA molecule is a potent technique that can be used to examine the antioxidant and pro-oxidant properties of a sample on cellular components [38].

The results showed that all of the tested extracts have antioxidant activity in the *E. coli* plasmid model used for the analysis. A lower level of DNA damage (less percentage of open form of plasmid DNA), however, was observed for the tested samples in comparison with the control which was treated only with hydrogen-peroxide and UV, and not with the extracts (Figure 1).

Interestingly enough, aqueous extracts showed better activity than the methanolic and ethanolic ones. IC_50_ values of aqueous extracts ranged from 160 µg/mL (for *Mp*) to 430 µg/mL (for *Ob*). Among all tested extracts, *Mp* methanolic, ethanolic, and aqueous extracts were found to have excellent genoprotective potential, followed by *So* and *Sm* extracts, while *Ob* was found to be the least potent genoprotective agent. The results obtained in this assay are moderately correlated with TPC, TFC, FC, caffeic acid, and DPPH (r ranged from −0.37 to −0.58); however, they are weakly correlated with PAC and the rest of the quantified phenolic compounds. Correlation coefficients have negative values due to presenting of the obtained results by IC_50_ values. The moderate correlation indicates that TPC (caffeic acid, in particular), TFC, and FC might partially be responsible for the displayed genoprotective activity of the tested extracts; however, further analysis is needed. Due to the fact that a moderate correlation was found between DPPH and PRA (r = −0.52), it can be concluded that the applied assays are compatible.

The literature data for genoprotective effect on plasmid DNA concerning plants from Lamiaceae family is lacking; however, the obtained results are of high significance since they highlighted a promising potential of different extracts for this activity.

### 2.4. Protective Effect of the Extracts against Hydrogen-Peroxide-Induced DNA Damage in the Prokaryotic Model

The protective effect of the extracts was primarily examined on *S. typhimurium* TA1535/pSK1002 since this assay has been recognized as a sensitive system for the detection of (anti)genotoxic agents [39]. Briefly, the principle of this assay is that if genotoxic lesions are produced in the bacterial genome, it will initiate SOS repair mechanism, with *umuC* gene expression being induced as part of the general SOS response. In this study, the level of SOS induction was measured by keeping track of the level of enzyme β-galactosidase [40]. The intensity of color development in the well is an indirect indicator of the produced β-galactosidase, which is further directly related to the amount of induced DNA damage. However, it should be once more highlighted that, in this study, the objective was to measure the antigenotoxic activity of the samples after an induced DNA damage. According to the obtained results, it can be concluded that most of the methanolic extracts significantly reduced DNA damage induced by hydrogen-peroxide. Among the ethanolic extracts, only *Sm* and *So* ones had an effect at the highest tested concentration (500 µg/mL), while none of the aqueous ones had significant impact on DNA damage (Figure 2). Among the methanolic extracts, *Ro* and *So* ones significantly reduced the DNA damage at all of the applied concentrations, and *Ob* only at 250 and 500 µg/mL, while the others (*Mo*, *Sm*, and ethanolic extracts of *So* and *Sm*) showed significant activity at 500 µg/mL only.

To the best of our knowledge, there is no existing literature data on the antigenotoxic effects of these particular plants and their extracts on *S. typhimurium* TA1535/pSK1002. However, from the results presented in this study it can be concluded that the tested extracts are eligible for further and more detailed investigation of their mechanisms of antigenotoxic action.

### 2.5. Protective Effect of the Extracts against Hydrogen-Peroxide-Induced DNA Damage in the Eukaryotic Model

#### 2.5.1. Determination of Cell Proliferation/Metabolic Viability

MTT assay was performed in order to determine the effect of the six Lamiaceae plant extracts on the viability of MRC-5 cells, which derived from normal lung tissue. It was found that the investigated plant extracts have no effect on the viability of these cells since in most cases their viability was found to be about 100% (data not shown). These findings are in accordance with the existing research data [41,42]. Since this cytotoxic effect of the extracts was not observed, the same concentrations of the extracts were further used for the assessment of the antigenotoxic effect in MRC-5 cells by comet assay.

#### 2.5.2. Comet Assay

The results of antigenotoxic potential of the tested Lamiaceae extracts evaluated using comet assay in co-treatments with hydrogen-peroxide performed for 2 h are shown in Figure 3. According to the obtained results, *Ob*, *Ro*, and *Sm* ethanolic extracts and aqueous extracts of *Mp* showed the best activity, where all of the tested concentrations significantly decreased the DNA damage. Moreover, the methanolic extracts significantly reduced the DNA damage at the lowest tested concentration in the case of *Mo*, *Mp*, and *So*.

Carvalho et al. [43] have found that only the ethanolic extracts of *Mo* significantly reduced the DNA damage, while aqueous extracts showed no activity. Ethanolic extracts of *Ob* were reported to possess the ability to neutralize the effects of hydrogen-peroxide in mammalian cells [44]. Razavi-Azarkhiavi et al. [45] reported that ethanolic extracts of *Ro*, in contrast to its aqueous ones, have DNA protecting activity. Furthermore, our results are in agreement with the study of Gateva et al. [46], where the treatment with water extract of *So* was shown to lead to a significant decrease in the DNA damage assessed by comet assay. Similarly, treatment of rat lymphocytes with the ethanolic extracts of *Satureja hortensis* resulted in a significant reduction of hydrogen peroxide-induced DNA damage compared to the control [47]; however, there is no available data on the effects of *Sm* extracts.

Previous research suggests that the antigenotoxic potential of these plants could be attributed to the presence of phenolic groups in their extracts [48,49]. It can be assumed that flavonols, such as quercetin, also have an effect on antigenotoxic activity, while large amounts of rutin found in plant extracts of the Lamiaceae family should not be neglected, as rutin has recently been shown to prevent UV-induced fibroblast cell membrane changes, while it also has a cytoprotective effect on cells exposed to various physical factors [50]. Thus, this metabolite could, among others, enable these extracts to prevent genotoxic damage inflicted on human DNA.

### 2.6. Antitumor Potential of the Extracts

The antitumor activity of the six Lamiaceae plants was assessed only for their ethanolic extracts since, according to our preliminary results, they were shown to be more active than their corresponding methanolic and aqueous extracts. The antitumor potential of ethanolic extracts was assessed using MTT, NBT, Griess, and comet assays.

#### 2.6.1. Determination of Antiproliferative/Cytotoxic Effect on Tumor Cells

The results of the MTT assay showed that only *Ob* and *Ro* ethanolic extracts have significantly decreased the proliferation of HCT-116 cells, and *Ob* at all concentrations, while *Ro* extract only at the lowest applied concentration (Figure 4).

Encalada et al. [51], on the other hand, have found that *Mo* ethanolic exert cytotoxicity on HCT-116 cells; however, this finding might be due to the application of a rather high concentration (1000 µg/mL) of extract, as opposed to a 10-fold lower concentration, which was used in our experiment. Although, in our literature survey, we did not come across any data regarding the effect of *Mp*, *Ob*, and *Ro* extracts on HCT-116 cells in MTT assay, Elansary et al. [52] reported that the methanolic extract of *Mp* did have an antiproliferative/cytotoxic effect on HT-29 cells (colon adenocarcinoma cell line). According to their study, this activity was also linked to the high amounts of rosmarinic acid and naringin present in the investigated *Mp* extract. Moreover, this activity was also reported for *Ob* and *Ro* methanolic extracts by Elansary et al. [33], and their results suggested that *Ro* extract had higher cytotoxic activity on HT-29 than *Ob* extract. The antiproliferative potential of *So* and *Sm* extracts on HCT-116 cells has not been extensively studied, as opposed to their essential oils, which were reported to have effect on different colon cancer cells (Caco-2, LS174, HT-29, and HCT-116) [6,53]. On the other hand, it is worth mentioning that one of the predominant phenolic compounds in these Lamiaceae species, rosmarinic acid, was found to inhibit the proliferation of HCT-116 cells [54].

Since some of the tested extracts had an antiproliferative effect on the colorectal cancer cell line, the next part of this study examined their effect on these cells to modulate the production of both ROS and NO.

#### 2.6.2. Determination of ROS-Modulating Activity in Tumor Cells

The results of NBT assay indicate that all of the investigated extracts significantly reduced the production of ROS by the HCT-116 cells, especially at their highest applied concentration, when compared with the untreated control (Figure 5). While moderate levels of ROS assist the cell proliferation and differentiation, according to a number of studies, tumor cells are prone to growth arrest or death when exposed to a higher concentration of ROS (so-called ROS-elevating approach), making them more sensitive to further oxidative stress [1,55]. However, there are indications that the role of ROS in tumor cells is two-sided, meaning that these cells might be more vulnerable in the absence of ROS (ROS-depleting approach) [56]. Since ROS might also function as proinflammatory mediator [57], decreased levels of ROS might reflect on the protection of normal, neighboring cells from excessive oxidative damage.

No similar results were earlier confirmed for these plants; however, it is worth mentioning that there are no studies investigating the effects of these plants’ extracts in the NBT assay. On the other hand, when using different methodology, it can be seen that these plants’ extracts and essential oils might lead to an increase in ROS production by colon cancer cells and, among others, to the inducing of apoptosis and inhibition of proliferation [58,59], which was proven as an effective approach in tumor therapy [1].

#### 2.6.3. Determination of NO-Modulating Activity in Tumor Cells

The NO-modulating activity was observed for all of the tested samples, significantly increasing the production of NO by HCT-116 cells when compared with the untreated control cells (Figure 6). Interestingly enough, it was observed that each extract had a higher effect when applied in the lowest concentration (25 µg/mL).

NO has an important homeostatic role in human organism [60]; however, studies indicate that NO represents a key signaling molecule with the ability to regulate the processes of tumorogenesis. As well as in the case of ROS, the role of NO in tumor cells appears to be controversial and not fully understood; however, numerous studies suggest that NO is cytotoxic to tumor cells and can inhibit tumor growth [61,62]. Bearing that in mind, the antitumor activity obtained in MTT assay can be explained through the increase in NO production by HCT-116 cells under the influence of the Lamiaceae extracts.

#### 2.6.4. Comet Assay

The ethanolic extracts tested in comet assay showed that *Mp*, *Ob*, and *Sm* inflict significant damage to the HCT-116 cells’ DNA, while *Ro* and *So* extracts did not exert a genotoxic effect on these cancer cells whatsoever. Interestingly enough, only *Ob* extract had similar effect to the one of etoposide, a well-known DNA-damaging agent (Figure 7).

To the best of our knowledge, there are no similar reports on the genotoxic effects of these six Lamiaceae species on HCT-116 cells. Nevertheless, other plants belonging to the Lamiaceae have been reported earlier to express genotoxic potential toward different tumor cells, such as colon adenocarcinoma, as well as breast and ovarian tumors [63,64]. Although no comparable studies were found, this research demonstrated that *Mp*, *Ob*, and *Sm* ethanolic extracts in the concentration of 100 µg/mL can cause DNA strand breaks in HCT-116 cells. Although further studies on their mechanism of antitumor action are needed, Blagosklonny [65] suggested that tumor cells divide faster than normal ones, and that those proliferating cells are more sensitive to the effects of adequate therapeutics. However, since antitumor agents are usually toxic to normal cells, it is preferable to develop therapeutics with the combined ability of protecting the normal cells and inducing damages to tumor cells. Actually, in this study, we have proven that *Ob* ethanolic extract not only has an antigenotoxic effect on normal cells, but it also showed significantly genotoxic activity on HCT-116 cells to the one of etoposide; hence, it is safe to assume that, besides the aforementioned effects, *Ob* might prevent the repair machinery to relegate the DNA strands, as etoposide does [66], which can have a crucial antitumor impact. Besides, tumor cells are more sensitive to external stimuli that can moderate the production of ROS [1]. Bearing that in mind, it can be concluded that natural substances with powerful antioxidant effects are more than able to act as potential antitumor therapeutics.

Finally, although in vitro studies are known to be quite informative and represent an invaluable experimental starting point, it is important to emphasize that the results obtained in vitro cannot fully explain the mechanisms that take place in the human body. More detailed studies on the bioavailability of these extracts are needed, as well as to enlighten their in vivo molecular mechanisms of antitumor activity because pH, temperature, and involvement of multiple regulators and other products of metabolism can interfere with their actions.

## 3. Materials and Methods

### 3.1. Chemicals and Reagents

All reagents and standards were of analytical, LC-MS, and HPLC grade. Methanol, ethanol, and hydrochloric acid were obtained from Zorka Pharma, Šabac, Serbia. Acridine orange, aluminium chloride (AlCl_3_), potassium acetate (CH_3_COOK), aluminum nitrate nonahydrate (Al(NO_3_)_3_ × 9H_2_O), ascorbic acid, bicarbonate buffer solution, caffeic acid, dimethyl sulfoxide (DMSO), EDTA, Folin-Ciocalteu reagent, gallic acid, low melting point agarose (LMP), MTT (3-(4,5-dimethylthiazol-2-yl)-2,5-diphenyltetrazolium bromide), N-(1-naphthyl) ethylenediamine dihydrochloride, normal melting point agarose (NMP), *ortho*-nitrophenyl-b-galactoside (ONPG), quercetin, sodium acetate (CH_3_COONa), sodium carbonate anhydrous (Na_2_CO_3_), TBE buffer, Tris, Triton X-100, trypsin, β-galactosidase, 1% sulphanilamide, 2,2-diphenyl-1-picrylhydrazyl (DPPH), 2-tert-butyl-4-hydroxyanisole (BHA), 3,5-di-tert-butyl-4-hydroxytoluene (BHT), and 4-nitroquinoline N-oxide (4NQO) were purchased from Sigma-Aldrich, St. Louis, MO, USA. Fetal bovine serum, 1×PBS (phosphate buffered saline solution), and Dulbecco’s Modified Eagle Medium (DMEM) were obtained from PAA Laboratories GmbH, Oberosterreich, Austria. Penicillin/streptomycin was purchased from PAA Laboratories, Austria. NBT (*p*-Nitrotetrazolium blue chloride) was obtained from Carl Roth, Karlsruhe, Germany. Sodium molybdate (Na_2_MoO_4_) and sodium nitrite (NaNO_2_) was purchased from Dispo-chem, Romsey, United Kingdom. Sodium chloride (NaCl) was purchased from Carlo Erba Reagents, Milano, Italy, while sodium hydroxide (NaOH) was purchased from Superlab, Belgrade, Serbia. Hydrogen peroxide (H_2_O_2_) was purchased from Farmanea Galenska Laboratorija, Belgrade, Serbia, while sodium hydroxide was purchased from NRK inženjering, Belgrade, Serbia.

### 3.2. Plant Material

Plant material was kindly obtained from the Institute for Medicinal Plant Research “Dr Josif Pančić”. *Mo*, *Mp*, *Ob*, *Ro*, and *So* were cultivated on the production fields of the Institute for Medicinal Plants Research “Dr Josif Pančić”, Pančevo, Serbia (N 44.872162, E 20.699931, 81 m a.s.l), while *Sm* was cultivated near villages Jelašnica and Prosek in South-Eastern Serbia (N 43.302149, E 22.061373, 293 m a.s.l). Aerial parts of these plants were collected in the spring of 2016, after which the material was dried at 50 °C in an industrial dryer, grounded, and stored afterwards in paper bags. Moreover, the specimens were deposited in the Herbarium of the Institute for Medicinal Plant Research “Dr Josif Pančić”, Belgrade (*Mo* voucher specimen number—301,121, *Mp*—301,131, *Ob*—302,061, *Ro*—301,211, *So*—301,241, and *Sm*—302,311).

Extracts were prepared as follows: 10 g of grinded plant material was subjected to the procedure of classic maceration (24 h, 25 °C) using 70% methanol, 70% ethanol, and hot distilled water as solvents. The mixtures (10% *w*/*v*) were exposed to ultrasound bath the first and the last hour of extraction (30 °C), the extracts were filtered twice (Whatman No. 1), and solvents were evaporated under reduced pressure with a rotary evaporator (Buchi Rotavapor R-114). The obtained crude extracts were stored in the refrigerator, protected from light and moisture.

### 3.3. Determination of Polyphenolic Components

First experiments regarding the determination of phytochemical contents of extracts were carried out within the first month of extracts preparation in order to avoid the possible degradation of their active compounds.

Total phenolic content (TPC) of the plants’ extracts was performed by the spectrophotometric procedure of Singleton and Rossi [67]. The reaction mixture was prepared by mixing 100 µL of extract (concentrations 100, 250, and 500 µg/mL) with 500 µL of 10% Folin-Ciocalteu’s reagent dissolved in water. After 6 min, 400 µL of 7.5% sodium carbonate was added. Blank contained distilled water instead of extracts. The absorbance was measured at 740 nm after 2 h incubation at room temperature, using a Perkin Elmer Lambda Bio UV/VIS spectrophotometer (Waltham, MA, USA). The same procedure was repeated for the standard solution of gallic acid (GA). Total phenolic content of the samples was calculated from the standard curve and expressed as mg GA equivalents per gram of dry extract, presented as means ± standard deviations.

The method for phenolic acids content (PAC) determination was done according to Mihailović et al. [68], with slight modifications. In each test tube 100 μL of extract (concentrations 100, 250, and 500 µg/mL), 200 μL of Arnow’s reagent (10% *w/v* sodium molybdate and 10% *w/v* sodium nitrite), 200 μL of 0.1 M chlorhydric acid, 200 μL of 1 M sodium hydroxide, and 1000 µL of distilled water were added. Blank contained each of the aforementioned components, while 50% ethanol was used instead of extracts. Caffeic acid (CA) was used for the construction of the calibration curve. The absorbance was measured at 490 nm using a Perkin Elmer Lambda Bio UV/VIS spectrophotometer (Waltham, MA, USA). The content of phenolic acids was expressed as mg CA equivalents per gram of dry extract, and the results were presented as means ± standard deviations.

Total flavonoid content (TFC) of the samples was measured spectrophotometrically according to the procedure given by Park et al. [69]. The reaction mixture was prepared by mixing 500 μL of extract (concentrations 100, 250, and 500 µg/mL) with 2005 μL of 80% ethanol, 50 μL of 10% aluminium nitrate nonahydrate, and 50 μL of 1 M potassium acetate. Blank contained 96% ethanol instead of extracts. After 40 min incubation at room temperature, the absorbance was measured at 415 nm using a Perkin Elmer Lambda Bio UV/VIS spectrophotometer (Waltham, MA, USA). The same procedure was repeated for quercetin (Q) in order to construct the calibration curve. The content of flavonoids in the samples was expressed as mg Q equivalents per gram of dry extract, as means ± standard deviations.

For the determination of flavonol content (FC), the procedure described by Mihailović et al. [68] was used with some modifications. In each test tube, 200 μL of extract (concentrations 100, 250, and 500 µg/mL) dissolved in 100% methanol, 200 μL of aluminium chloride solution (20 mg/mL), and 600 μL of sodium acetate (50 mg/mL) methanol solution were added. Blank contained all of the mentioned components, and 100% methanol instead of extracts. For the construction of the calibration curve, Q was added instead of extracts. The absorbance was measured at 440 nm after 150 min of incubation, using a Perkin Elmer Lambda Bio UV/VIS spectrophotometer (Waltham, MA, USA). Flavonol content was calculated from the calibration curve equation and expressed as mg Q equivalents per gram of dry extract. The results are presented as means ± standard deviations.

### 3.4. HPLC-DAD Quantification of Polyphenols

The polyphenols in the tested samples were quantified by comparing the retention times and absorption spectra (200–400 nm) of unknown peaks with the reference standards. HPLC-DAD analysis was performed on Agilent 1200 Series HPLC (Agilent Technologies, Palo Alto, CA, USA) equipped with Lichrospher^®^ 100 RP 18e column (5 μm, 250 × 4 mm). Aqueous solution of formic acid (0.17%) was used as mobile phase A, while acetonitrile served as the mobile phase B. The injection volume was 10 μL, and flow rate was 0.8 mL/min with gradient program (0–53 min 0–100% B). The analysis was stopped after 55 min. The results were obtained by comparing the retention times and absorption spectra (200–400 nm) of unknown peaks with the reference standards. The investigated samples were analyzed in triplicate.

### 3.5. Evaluation of DPPH-Scavenging Activity

For the evaluation of the antioxidant activity of extracts, DPPH free radical scavenging method [70] was used with slight modifications. In each test tube, 100 μL of extract (concentrations 100, 250, and 500 µg/mL) and 900 µL of methanolic solution of DPPH (40 μg/mL) were added. Methanol was used as blank, and methanol with DPPH solution was used as negative control, while BHA, BHT, and ascorbic acid were used as positive controls (standards). Absorbance was measured at 517 nm after 30 min in the dark at room temperature, using a Perkin Elmer Lambda Bio UV/VIS spectrophotometer (Waltham, MA, USA). The decrease in absorption of DPPH was calculated as follows:Inhibition of DPPH radical (%) = (A_c_ − A_s_)/A_c_ × 100,
where A_c_ represents the absorbance of the negative control, and A_s_ is the absorbance of the test samples. The results are presented as percentage of DPPH inhibition ± standard deviations.

### 3.6. Genoprotective Activity in Acellular System

The effects of extracts on the protection of supercoiled DNA were studied against OH radical (generated by the photolysis of hydrogen peroxide), in an acellular system using plasmid relaxation assay (PRA), described by Russo et al. [37]. Plasmid pUC19 was isolated from *E. coli* XL1-blue following the protocol by Birnboim [71]. The isolated plasmid was checked after 1 h electrophoresis on 1% agarose gel (0.5 × TBE, 80 V, 300 mA). About 80% of the isolated plasmid was present in the closed circular form, while about 20% was in the open circular (nicked) form.

All experiments were performed in a volume of 15 µL of 1 × PBS containing 800 ng of plasmid pUC19 and plant extracts in the final concentrations of 100, 500, and 1000 µg/mL. Solvent controls were included in experiments where the extracts were substituted with ethanol or methanol (final concentration in mixture was 7%). Immediately prior to the irradiation of the samples with UV light, hydrogen peroxide was added in a final concentration of 3 mM. The reaction volumes were held in cover of 96-well plate, placed directly under the UV lamp and irradiated for 3 min (540 J/m^2^) at 254 nm. The interaction of extracts with the plasmid DNA was interrupted by the addition of 5 µL of 6 × loading dye (30% (*v*/*v*) glycerol, 0.25% (*w*/*v*) bromophenol blue, and 0.25% (*w*/*v*) xylene cyanol). UV irradiated (540 J/m^2^) mixture of 3 mM hydrogen peroxide, and plasmid served as a positive control. The samples were analyzed by electrophoresis on a 1% agarose gel prepared in 0.5×TBE buffer. The positions of open circular form and closed circular form of plasmid were compared to the ladder λ/PstI (0.5 µg/lane, Thermo Fisher Scientific, Waltham, MA, USA). Electrophoresis was performed at a constant voltage of 80 V, 300 mA for about 1 h, whereas bromphenol blue did not exceed 75% of the gel. Gels were observed on a UV transilluminator under an ultraviolet light at 312 nm and photographed with a digital camera. The images were analyzed using software ImageJ (National Institutes of Health, Bethesda, MD, USA). Treatments were performed in three individual experiments.

### 3.7. Protective Effect of the Extracts against Hydrogen Peroxide-Induced DNA Damage in the Prokaryotic Model

The inhibition of DNA damage induced by hydrogen-peroxide in co-treatment with selected extracts was assessed in SOS/*umuC* assay, following the protocol given in Kolarević et al. [39]. A series of preliminary treatments with hydrogen peroxide were performed and the concentration of 1.68 mM was chosen as the most appropriate for further testing (high induction rate and low growth inhibition; data not shown). The treatment with hydrogen peroxide in the final concentration of 1.68 mM was performed for 2 h at 37 °C in incubation mixtures composed of 10 µL of extract (concentrations 125, 250, and 500 µg/mL) and 90 µL of bacterial culture of *Salmonella typhimurium* TA1535/pSK1002 in the exponential phase. Sterile bidistilled water was used as negative control, while methanol or ethanol in final concentration of 3.5% served as solvent controls. After incubation, mixtures were diluted 10 times, incubated for 2 h, and the bacterial growth rate was determined by measuring the absorbance at 600 nm on a Multiskan FC microtiter plate reader (Thermo Fisher Scientific, Waltham, MA, USA). β-galactosidase activity (G) was determined using ONPG as a substrate (30 min at 25 °C). The absorption was measured at 405 nm using reference solution without bacteria. The bacterial growth rate was calculated using the following formula:G = sample_OD600_/control_OD600_.

A growth ratio less than 0.75, which represents 25% inhibition of biomass, was considered to be an indication of cytotoxicity. Induction rate (IR) was calculated as follows:IR = sample_OD405_/control_OD405_ × G.

All treatments were performed in triplicate in three individual experiments.

### 3.8. The Effects of Extracts on Normal and Tumor Cell Lines

#### 3.8.1. Cell Culture

Human fetal lung fibroblast, MRC-5, as well as human colon cancer HCT-116 cell lines (obtained from the American Tissue Culture Collection, ATCC, Manassas, VA, USA), were grown in the cultivation medium (DMEM supplemented with fetal bovine serum in final concentration 10%, and 100 U/mL penicillin/streptomycin) at 37 °C in a humidified atmosphere containing 5% CO_2_.

#### 3.8.2. Determination of Cell Proliferation/Metabolic Viability (MTT Assay)

MTT test was performed following the protocol described in detail in Kolarević et al. [39]. The extracts were tested in three concentrations (25, 50, and 100 µg/mL). Solvent controls consisted of ethanol or methanol in a final concentration 0.35%. Following the treatment, medium was removed, and the MRC-5 and HCT-116 monolayer was further washed with 1×PBS. To each well 200 µL of MTT solution (500 µg/mL) was added. The plates were then incubated for 3 h at 37 °C, 5% CO_2_. The supernatant was extracted from the wells, and formazan crystals were dissolved by adding 200 µL SDS-HCl (10% SDS in 0.1% 1N hydrochloric acid) per well. Finally, the absorbance was measured at reduced MTT was assayed at 540 nm using a Multiskan FC microtiter plate reader (Thermo Fisher Scientific, Waltham, MA, USA). The results are expressed as the percentage of viable cells, calculated as the ratio between the absorbance of treated cells and the absorbance of the untreated control multiplied by 100.

#### 3.8.3. Comet Assay

The antigenotoxic effect of extracts against hydrogen peroxide-induced DNA damage assessed on normal cells, as well as the genotoxic effects assessed in tumor cells, was based on the study of Yang et al. [72]. MRC-5 and HCT-116 cells were seeded in 24-well plates (0.7 mL/well) in a concentration of 10^5^ cells/mL and left for 24 h at 37 °C to attach. For the normal cells, the cultivation medium was replaced with DMEM containing 50 µM hydrogen peroxide and different concentrations of extracts, while the tumor cells were only treated with the ethanolic extracts. The concentrations of extracts were selected based on the results of the MTT assay (25, 50, and 100 µg/mL); however, on HCT-116 cells, only the ethanolic extracts at the concentration of 100 µg/mL were tested. For solvent controls, instead of extracts, methanol and ethanol were used in a final concentration of 0.35%. Treatments were performed during 60–120 min. At the end of this incubation period, cells were trypsinized, centrifuged for 10 min at 1000 rpm, and eluted in 1 × PBS to obtain approximately 10^5^ cells/mL.

Comet assay was performed in minigel format as described in Azqueta et al. [73] with some modifications. Aliquots of cell suspensions obtained after treatment (30 µL) were mixed with 70 µL of 1% low melting point agarose. For each sample, 15 µL of the prepared mix was placed on slides pre-coated with 1% normal melting point agarose. Each microscope slide contained 10 minigels: negative controls, three different concentrations of extracts indicated above and positive controls (hydrogen peroxide), all in duplicate. The slides were incubated for 5 min at 4 °C and lysed for 1 h in freshly prepared, cold (4 °C) lysis buffer (2.5 M NaCl, 100 mM EDTA, 10 mM Tris, 10% DMSO, 1.5% Triton X-100, pH 10) and then transferred to an electrophoresis chamber containing cold (4 °C) alkaline electrophoresis buffer (300 mM NaOH, 1 mM EDTA, pH 13). The first step, the denaturation of DNA, involved the incubation of slides for 20 min at 4 °C without electricity, after which the current was released (0.75 V/cm, 300 mA) for 20 min. After electrophoresis, the slides were transferred to a freshly cold (4 °C) neutralizing buffer (0.4 M Tris, pH 7.5) for 15 min. The slides were then fixed in ethanol at 4 °C for 15 min and air dried for 24 h. For visualization, the minigels were stained with 20 µL of acridine orange (2 µg/mL) per slide. Fifty nucleoids per minigel (a total of 100 nucleoids per sample) were scored by fluorescent microscope (Leica, DMLS, Vienna, Austria, under magnification 400×, excitation filter 450–490 nm, barrier filter 510 nm) using Comet IV Computer Software (Perceptive Instruments, Bury St. Edmunds, UK) software in order to obtain values for Tail Intensity parameter (TI%) (Figure 8).

#### 3.8.4. Determination of Superoxide Anion Radical (NBT Assay)

The concentration of superoxide anion radical (O_2_^−^) in the samples was determined by the NBT assay [74]. HCT-116 cells were incubated for 24 h with the ethanolic extracts, followed by the removal of 100 µL of medium, the addition of 10 µL of NBT solution (5 mg/mL in PBS), and the incubation of the cells for 3 h at 37 °C in 5% CO_2_. After that, in order to quantify the production of formazan, the cells were solubilized in 100 µL SDS-HCl (10% SDS in 0.1% 1 N hydrochloric acid). The results are expressed as NBT index, calculated as the ratio between the absorbance of treated cells and the absorbance of the untreated control.

#### 3.8.5. Determination of the Levels of Nitrites in Supernatants (Griess Assay)

The determination of nitrites (NO_2_^−^) was performed using the modified spectrophotometric method first described by Griess et al. [75]. Griess reagent consisted of component A (N-(1-naphthyl) ethylenediamine dihydrochloride in distilled water) and component B (1% sulfanilamide in 5% phosphoric acid) mixed together in equal volumes immediately prior to the application on the plate. After a 24-h incubation of the cells with the ethanolic extracts, 50 µL of medium from each well was transferred to an empty microplate followed by the addition of the prepared Greiss reagent (50 µL) to each well. After 10 min in the dark, the absorbance was measured at 540 nm using a microplate reader (LKB 5060–006, LKB Instruments, Vienna, Austria). The concentration of nitrites was calculated from the standard curve for nitrite and expressed in µmol/L (µM).

### 3.9. Statistical Analyses

Statistical analyses were carried out using Statistica Software. Kolmogorov-Smirnoff was used to determine if the data were normally distributed. As the data were in line for the usage of nonparametric tests, Kruskal–Wallis one-way ANOVA was applied followed by Mann–Whitney U test for pairwise comparison of the treated groups with negative and solvent controls. Additional statistical evaluation was carried out by Independent Samples *t*-test. The level of significance for all comparisons was set at *p* < 0.05. Pearson’s correlation coefficients (r) were calculated among results of the DPPH assay, PRA, and phytochemical characterization and presented according to Taylor [34]. For the SOS/*umuC* assay, the significance of DNA damage inhibition in co-treatments with extracts in comparison with hydrogen peroxide alone was assessed with a *t*-test.

## 4. Conclusions

This study demonstrated and confirmed the potential of aromatic plants as excellent sources of health-boosting compounds. The obtained data showed that methanolic extracts were generally the richest in PAC, TFC, and FC, and they also exerted the highest DPPH-scavenging activity and protective potential in SOS/*umuC* assay. Ethanolic extracts had the highest TPC, and they also exhibited the highest antigenotoxic effects in the comet test. Additionally, rosmarinic acid, quercetin, rutin, and luteolin-7-*O*-glucoside were among the identified compounds found in the investigated Lamiaceae samples. On the other hand, genoprotective effect on plasmid DNA was obtained for all of the tested extracts, with the best results for aqueous extracts. A moderate to strong correlation was found between phytochemical content and antioxidant activity both in vitro and on an acellular model. Overall, in this study, *Melissa officinalis* and *Mentha* × *piperita* proved to be the most active genoprotective agents. On the other hand, regarding the antitumor activity of these plants, *Ocimum basilicum* ethanolic extract stood out as the one with the highest potential for further research since it also had a noticeable antigenotoxic effect on DNA of normal human cells. Finally, the obtained results represent a novelty in the field of testing the genoprotective and antigenotoxic effects of plant extracts from the Lamiaceae family. Moreover, to the best of our knowledge, this is the first comprehensive and comparative study regarding these six Lamiaceae species and their overall genoprotective and antitumor activities. Hence, it can be expected that the presented findings will foster further studies of these potential nutraceutical agents leading to the development of a valuable database with special importance for food and pharmaceutical industries.

## Figures and Tables

**Figure 1 plants-10-02306-f001:**
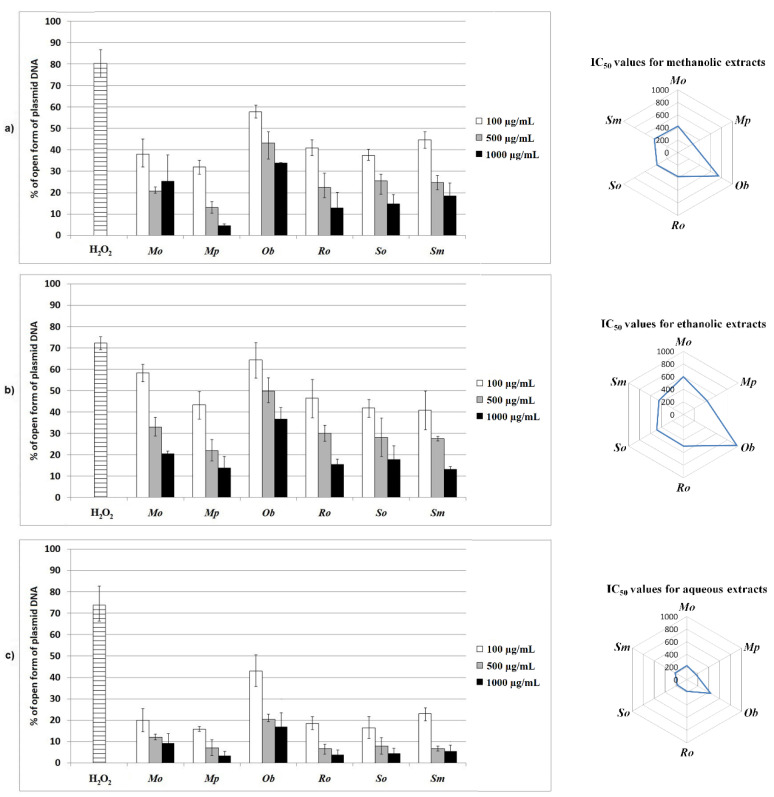
Percentage (%) of DNA damage (open form of plasmid DNA) in proportion to the concentrations of extracts (100, 500, and 1000 µg/mL) and their IC_50_ values (µg/mL). (**a**) Methanolic extracts; (**b**) ethanolic extracts; (**c**) aqueous extracts. H_2_O_2_—positive control. The level of statistical significance was set at *p* < 0.05.

**Figure 2 plants-10-02306-f002:**
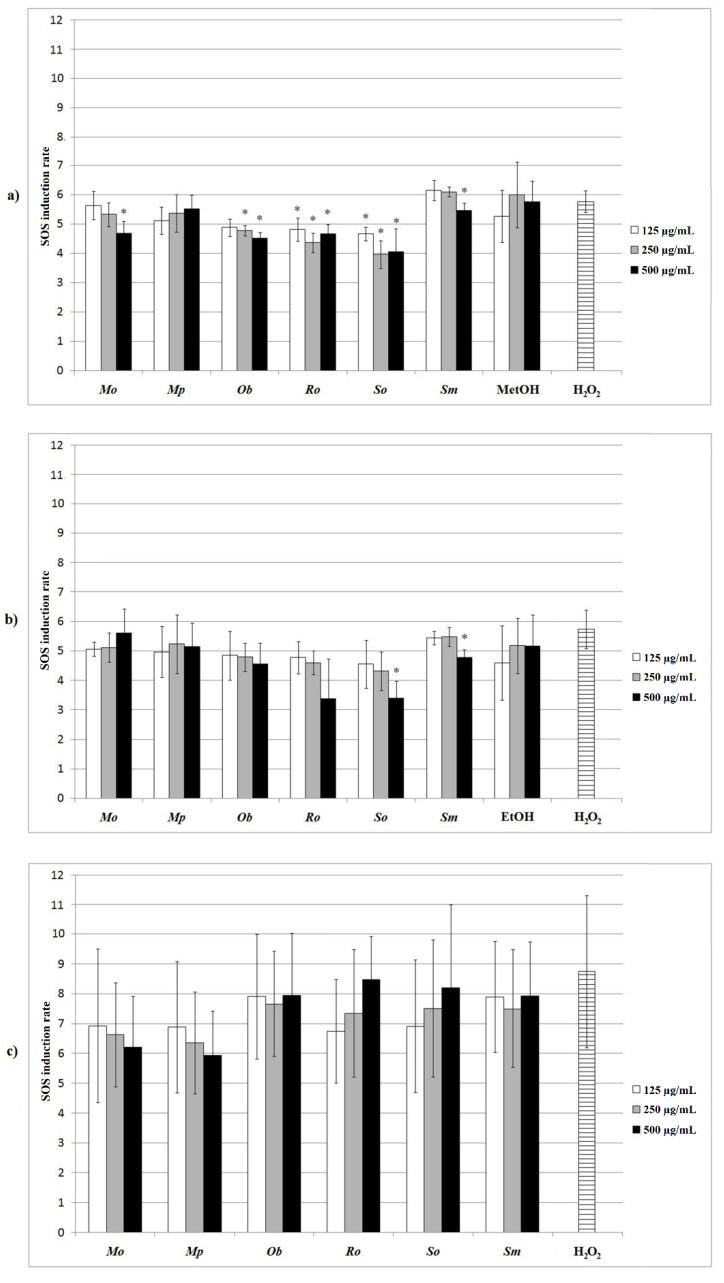
The impact of (**a**) methanolic extracts; (**b**) ethanolic extracts; (**c**) aqueous extracts, at 125, 250, and 500 µg/mL on the induction ratio of SOS response in SOS/*umuC* test. MetOH and EtOH—controls for methanol and ethanol, respectively.* *p* < 0.05—statistically significant decrease of DNA damage in comparison with the negative control (H_2_O_2_).

**Figure 3 plants-10-02306-f003:**
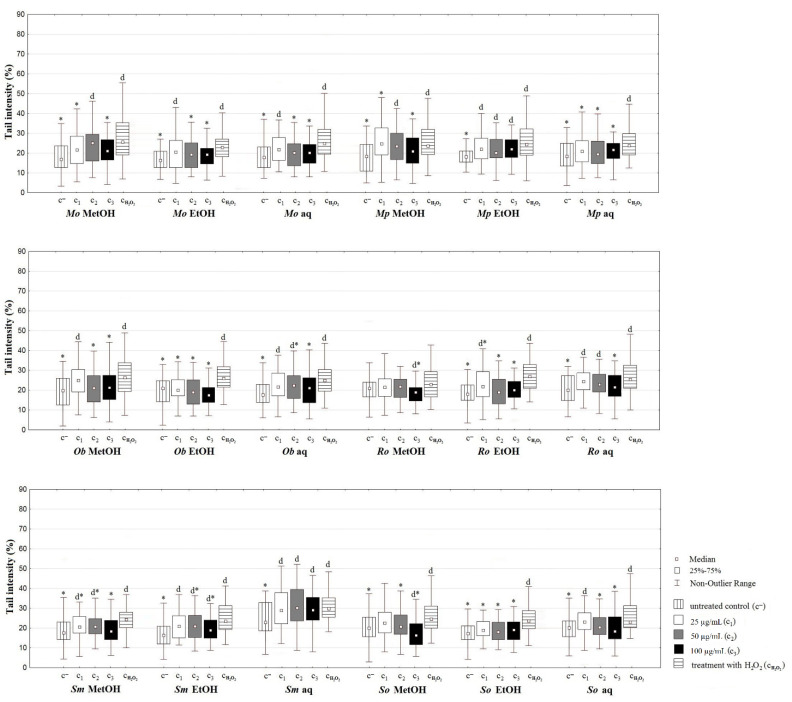
Comet assay performed on the MRC-5 cell line with the extracts’ concentrations of 25, 50, and 100 µg/mL. MetOH, EtOH, aq—methanol, ethanol, and aqueous extracts, respectively. * *p* < 0.05—statistically significant decrease of DNA damage in comparison with the treated control (c_H2O2_); d *p* < 0.05—statistically significant increase of DNA damage in comparison with the untreated control (c^−^).

**Figure 4 plants-10-02306-f004:**
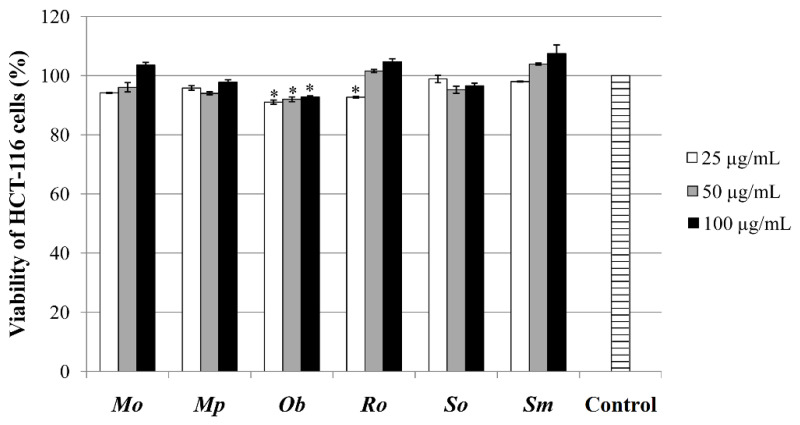
MTT assay performed in the extracts’ concentrations of 25, 50, and 100 µg/mL. *—statistically significant decrease in proliferation of HCT-116 cells in comparison with the untreated control, *p* < 0.05.

**Figure 5 plants-10-02306-f005:**
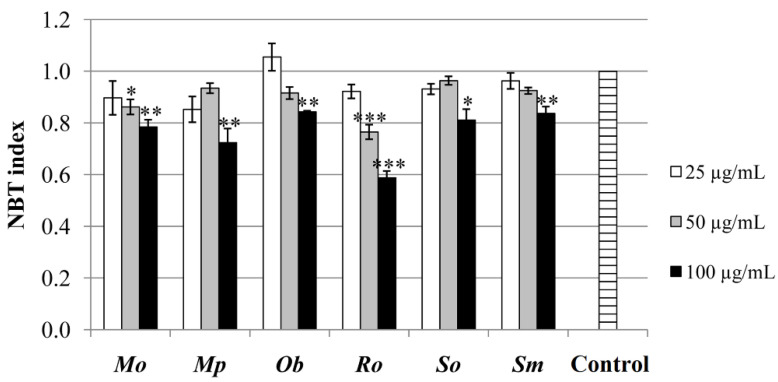
NBT assay performed in the extracts’ concentrations of 25, 50, and 100 µg/mL and expressed as NBT index–ratio between absorbance of treated cells and untreated control, calculated on 100% viable cells. * *p* < 0.05; ** *p* < 0.01; *** *p* < 0.001—statistically significant increase in ROS production by HCT-116 cells in comparison with the untreated control.

**Figure 6 plants-10-02306-f006:**
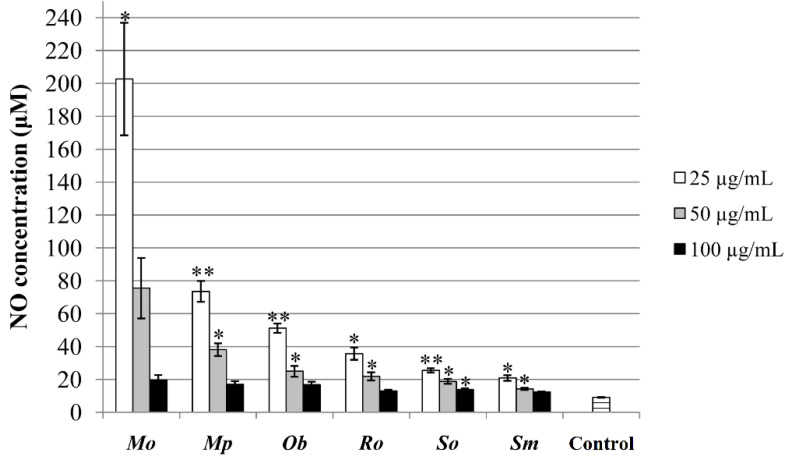
Griess assay performed in the extracts’ concentrations of 25, 50, and 100 µg/mL and expressed as the concentration of nitric oxide (NO). * *p* < 0.05; ** *p* < 0.01—statistically significant increase in NO production by HCT-116 cells in comparison with the untreated control.

**Figure 7 plants-10-02306-f007:**
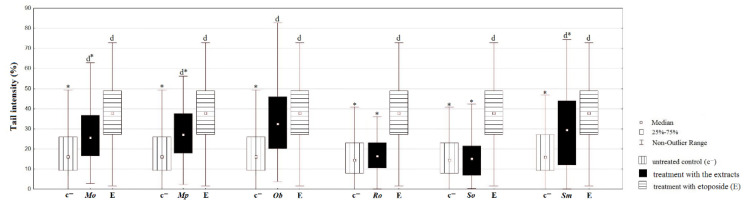
Comet assay performed on HCT-116 cell line with the ethanolic extracts at the concentration of 100 µg/mL and expressed as percentage of tail intensity. * *p* < 0.05—significantly lower DNA damage in comparison with etoposide (E); d *p* < 0.05—statistically significant increase of DNA damage in comparison with the untreated control, c^−^.

**Figure 8 plants-10-02306-f008:**
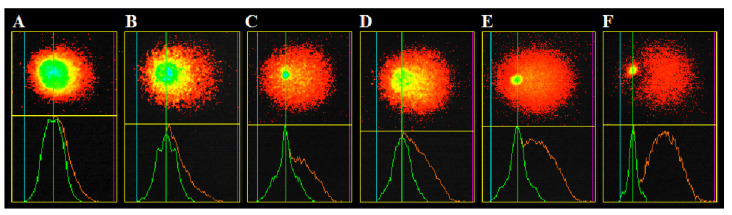
Representative images of nucleoids scored in comet assay, with different levels of TI%: (**A**)—18.97%, (**B**)—29.94%, (**C**)—38.08%, (**D**)—46.58%, (**E**)—56.74%, and (**F**)—78.28%.

**Table 1 plants-10-02306-t001:** The content of polyphenolic compounds (total phenolic (TPC), phenolic acid (PAC), total flavonoid (TFC), and flavonol (FC) contents) of the selected plant extracts presented at 250 µg/mL, as means ± standard deviation.

		The Content of Polyphenolic Compounds			
Species	Extract	TPC (mg GAE/g)	PAC (mg CAE/g)	TFC (mg QE/g)	FC (mg QE/g)
*Mo*	Methanolic	79.40 ± 3.78 ^b^	58.26 ± 8.34 ^c^	11.25 ± 1.13 ^c^	0.90 ± 0.29
	Ethanolic	70.30 ± 1.52 ^a^	67.89 ± 1.92 ^c^	10.13 ± 0.43 ^c^	0.96 ± 0.23
	Aqueous	74.22 ± 1.39	40.48 ± 2.31 ^ab^	5.97 ± 0.22 ^ab^	1.30 ± 0.55
*Mp*	Methanolic	77.12 ± 4.11	51.59 ± 2.80 ^c^	32.00 ± 0.44 ^c^	11.09 ± 0.20 ^c^
	Ethanolic	70.79 ± 2.10	54.56 ± 2.94 ^c^	30.85 ± 0.35 ^c^	11.34 ± 0.26 ^c^
	Aqueous	76.27 ± 2.64	16.41 ± 1.70 ^ab^	21.31 ± 0.34 ^ab^	5.33 ± 0.21 ^ab^
*Ob*	Methanolic	19.71 ± 0.40 ^b^	12.70 ± 7.40 ^c^	8.58 ± 0.35 ^b^	0.96 ± 0.47 ^b^
	Ethanolic	29.14 ± 0.37 ^a^	19.00 ± 3.33 ^c^	10.28 ± 0.33 ^a^	2.67 ± 0.21 ^ac^
	Aqueous	24.28 ± 0.17	nd ^ab^	8.76 ± 0.57	0.55 ± 0.54 ^b^
*Ro*	Methanolic	57.32 ± 0.54 ^c^	68.63 ± 8.34 ^bc^	18.52 ± 1.15 ^bc^	3.75 ± 0.13 ^bc^
	Ethanolic	54.09 ± 2.33 ^c^	33.07 ± 1.70 ^a^	12.09 ± 0.28 ^ac^	2.50 ± 0.06 ^ac^
	Aqueous	76.37 ± 0.74 ^ab^	33.44 ± 4.44 ^a^	14.82 ± 0.38 ^ab^	5.77 ± 0.43 ^ab^
*So*	Methanolic	70.79 ± 2.16 ^c^	46.78 ± 6.19	26.93 ± 0.67 ^ac^	8.84 ± 0.39 ^b^
	Ethanolic	69.26 ± 5.77 ^c^	50.11 ± 10.94 ^c^	19.95 ± 0.05 ^a^	6.91 ± 0.20 ^ac^
	Aqueous	44.63 ± 0.64 ^ab^	30.85 ± 2.31 ^b^	20.50 ± 0.14 ^a^	8.01 ± 0.30 ^b^
*Sm*	Methanolic	55.17 ± 1.87 ^bc^	64.56 ± 3.33 ^bc^	10.47 ± 0.53 ^bc^	4.39 ± 0.27 ^c^
	Ethanolic	66.52 ± 1.21 ^a^	44.93 ± 3.57 ^a^	12.30 ± 0.22 ^a^	4.06 ± 0.31 ^c^
	Aqueous	63.49 ± 1.57 ^a^	35.30 ± 7.23 ^a^	12.46 ± 0.24 ^a^	6.62 ± 0.29 ^ab^

The following coefficients were used to highlight the statistical difference (*p* < 0.05) between the extracts of the same plant species: ^a^—vs. methanolic extract, ^b^—vs. ethanolic extract, ^c^—vs. aqueous extract. nd—not detected.

**Table 2 plants-10-02306-t002:** Chemical analysis of the extracts assessed using HPLC-DAD technique. The results are presented as mg/g dry extract, as means ± standard deviation.

Spices	Extract	Rosmarinic Acid	Caffeic Acid	Chlorogenic Acid	Quercetin	Rutin	Naringin	Luteolin-7-*O*-glucoside
*Mo*	Methanolic	86.76 ± 4.69 ^bc^	1.33 ± 0.08	0.23 ± 0.01 ^b^	12.43 ± 0.48 ^c^	1.64 ± 0.05 ^bc^	3.36 ± 0.13 ^bc^	2.55 ± 0.15
	Ethanolic	61.77 ± 3.58 ^ac^	2.54 ± 0.10	1.11 ± 0.04 ^ac^	11.56 ± 0.59 ^c^	3.38 ± 0.15 ^a^	2.94 ± 0.11 ^ac^	1.34 ± 0.10
	Aqueous	52.32 ± 3.18 ^ab^	2.87 ± 0.14	0.60 ± 0.01 ^b^	7.75 ± 0.35 ^ab^	3.40 ± 0.18 ^a^	1.44 ± 0.07 ^ba^	1.64 ± 0.08
*Mp*	Methanolic	63.30 ± 3.21 ^bc^	tr	0.35 ± 0.01 ^c^	2.94 ± 0.17 ^c^	tr ^bc^	2.42 ± 0.09 ^b^	5.07 ± 0.28
	Ethanolic	3.81 ± 1.15 ^ac^	tr	0.22 ± 0.01 ^c^	3.36 ± 0.15 ^c^	0.56 ± 0.29 ^ac^	1.20 ± 0.05 ^ac^	5.09 ± 0.29
	Aqueous	84.45 ± 4.15 ^ab^	0.69 ± 0.02	1.09 ± 0.05 ^ab^	11.05 ± 0.42 ^ab^	6.40 ± 0.31 ^ab^	2.41 ± 0.10 ^b^	4.41 ± 0.18
*Ob*	Methanolic	13.85 ± 0.42 ^c^	1.33 ± 0.03	0.83 ± 0.01 ^c^	3.03 ± 0.05 ^c^	2.70 ± 0.08 ^c^	3.26 ± 0.09 ^c^	1.25 ± 0.02
	Ethanolic	12.02 ± 0.28 ^c^	tr	0.62 ± 0.01 ^c^	2.62 ± 0.09	3.09 ± 0.10 ^c^	3.15 ± 0.18 ^c^	0.99 ± 0.01
	Aqueous	4.36 ± 0.11 ^ab^	0.78 ± 0.02	tr ^ab^	2.19 ± 0.07 ^a^	1.97 ± 0.05 ^ab^	1.83 ± 0.13 ^ab^	1.11 ± 0.02
*Ro*	Methanolic	63.48 ± 3.87 ^bc^	2.14 ± 0.09 ^b^	4.75 ± 0.27 ^bc^	tr^b^	10.04 ± 0.58 ^b^	1.08 ± 0.05	8.04 ± 0.46
	Ethanolic	49.44 ± 2.90 ^a^	0.08 ± 0.00 ^ac^	0.06 ± 0.00 ^ac^	1.95 ± 0.05 ^ac^	8.09 ± 0.32 ^ac^	0.92 ± 0.03	8.46 ± 0.42 ^c^
	Aqueous	45.87 ± 2.64 ^a^	2.40 ± 0.11 ^b^	2.61 ± 0.13 ^ab^	tr^b^	10.84 ± 0.48 ^b^	1.10 ± 0.03	7.02 ± 0.31 ^b^
*So*	Methanolic	55.62 ± 2.52 ^bc^	1.97 ± 0.06 ^c^	0.17 ± 0.00 ^bc^	3.28 ± 0.23 ^b^	8.82 ± 0.31 ^bc^	12.83 ± 0.56 ^bc^	22.82 ± 1.05 ^bc^
	Ethanolic	42.35 ± 2.66 ^ac^	2.38 ± 0.09 ^c^	0.60 ± 0.02 ^ac^	2.35 ± 0.10 ^a^	tr ^a^	9.88 ± 0.38 ^a^	18.65 ± 0.53 ^ac^
	Aqueous	29.01 ± 1.13 ^ab^	3.47 ± 0.13 ^ab^	1.39 ± 0.06 ^ab^	2.58 ± 0.11	tr ^a^	9.84 ± 0.41 ^a^	16.97 ± 0.69 ^ab^
*Sm*	Methanolic	51.96 ± 2.41 ^bc^	1.61 ± 0.05 ^bc^	tr ^bc^	3.37 ± 0.14 ^bc^	1.65 ± 0.07 ^bc^	1.62 ± 0.09	5.18 ± 0.19 ^bc^
	Ethanolic	39.60 ± 1.78 ^ac^	2.79 ± 0.09 ^ac^	0.79 ± 0.03 ^a^	2.03 ± 0.10 ^a^	tr^a^	1.48 ± 0.06	6.97 ± 0.32 ^a^
	Aqueous	13.44 ± 0.62 ^ab^	4.02 ± 0.19 ^ab^	0.66 ± 0.02 ^a^	2.22 ± 0.08 ^a^	tr^a^	1.54 ± 0.40	7.00 ± 0.38 ^a^

The following coefficients were used to highlight the statistical difference (*p* < 0.05) between the extracts of the same plant species: ^a^—vs. methanolic extract, ^b^—vs. ethanolic extract, ^c^—vs. aqueous extract, tr—traces.

**Table 3 plants-10-02306-t003:** Correlation coefficients between the assays used to analyze the phytochemical content (TPC, PAC, TFC, and FC), quantification of chemical compounds (RA—rosmarinic acid, CA—caffeic acid, ChA—chlorogenic acid, Q—quercetin, RUT—rutin, NAR—naringin, L-G—luteolin-7-*O*-glucoside), antioxidant activity in vitro (DPPH assay), and genoprotective activity in an acellular model (PRA—Plasmid Relaxation Assay).

	TPC	PAC	TFC	FC	RA	CA	ChA	Q	RUT	NAR	L-G	DPPH	PRA
TPC	1	0.90 ^c^	0.78 ^c^	0.69 ^c^	0.68 ^c^	0.22 ^a^	−0.06 ^a^	0.47 ^b^	0.28 ^a^	0.00 ^a^	0.21 ^a^	0.89 ^c^	−0.58 ^b^
PAC		1	0.65 ^b^	0.56 ^b^	0.60 ^b^	0.01 ^a^	−0.04 ^a^	0.48 ^b^	0.06 ^a^	−0.16 ^a^	0.04 ^a^	0.79 ^c^	−0.33 ^a^
TFC			1	0.93 ^c^	0.15 ^a^	0.11 ^a^	0.02 ^a^	−0.22 ^a^	0.33 ^a^	0.40 ^b^	0.59 ^b^	0.75 ^c^	−0.38 ^b^
FC				1	−0.08 ^a^	0.46 ^b^	−0.09 ^a^	−0.33 ^a^	0.15 ^a^	0.26 ^a^	0.47 ^b^	0.65 ^b^	−0.38 ^b^
RA					1	−0.26 ^a^	0.16 ^a^	0.70 ^c^	0.37 ^b^	0.05 ^a^	0.08 ^a^	0.46 ^b^	−0.34 ^a^
CA						1	0.15 ^a^	−0.15 ^a^	0.10 ^a^	0.21 ^a^	0.26 ^a^	0.58 ^b^	−0.37 ^b^
ChA							1	0.20 ^a^	0.59 ^b^	−0.21 ^a^	−0.01 ^a^	−0.02 ^a^	−0.19 ^a^
Q								1	−0.07 ^a^	−0.14 ^a^	−0.36 ^a^	0.25 ^a^	−0.15 ^a^
RUT									1	0.19 ^a^	0.61 ^b^	0.12 ^a^	−0.39 ^b^
NAR										1	0.83 ^c^	0.09 ^a^	−0.02 ^a^
L-G											1	0.22 ^a^	−0.29 ^a^
DPPH												1	−0.52 ^b^
PRA													1

According to Taylor (1990): ^a^ r ≤ 0.35 weak correlation; ^b^ 0.36 < r < 0.67 moderate correlation; ^c^ 0.68 < r < 1 strong correlation.

**Table 4 plants-10-02306-t004:** The activity of the selected plant extracts and standards (positive controls) against DPPH radical, presented as means of percentage of inhibition ± standard deviation.

DPPH Assay (Percentage of Inhibition)				
Species	Extract	100 µg/mL	250 µg/mL	500 µg/mL
*Mo*	Methanolic	41.81 ± 1.70 ^c, xyz^	90.50 ± 1.04 ^xy^	93.08 ± 0.29 ^xy^
	Ethanolic	38.47 ± 0.52 ^c, yz^	93.29 ± 0.13 ^xy^	93.58 ± 0.29 ^xy^
	Aqueous	48.19 ± 0.72 ^ab, xyz^	92.42 ± 0.68 ^xy^	92.36 ± 0.67 ^xy^
*Mp*	Methanolic	55.86 ± 1.08 ^bc, xyz^	93.40 ± 0.77 ^c, xy^	93.33 ± 0.34 ^xy^
	Ethanolic	40.09 ± 0.67 ^ac, yz^	92.80 ± 0.64 ^c, xy^	94.46 ± 0.28 ^c, xy^
	Aqueous	23.88 ± 0.66 ^ab, xyz^	61.27 ± 2.84 ^ab, xyz^	91.99 ± 0.38 ^b, xy^
*Ob*	Methanolic	15.84 ± 0.94 ^xyz^	42.50 ± 1.70 ^bc, xyz^	69.95 ± 0.56 ^bc, xz^
	Ethanolic	14.05 ± 1.35 ^xyz^	34.43 ± 0.36 ^ac, xyz^	54.02 ± 0.90 ^ac, xyz^
	Aqueous	11.89 ± 1.12 ^xyz^	27.68 ± 0.07 ^ab, xyz^	51.61 ± 0.24 ^ab, xyz^
*Ro*	Methanolic	37.25 ± 0.51 ^byz^	64.14 ± 0.41 ^bc, xyz^	94.98 ± 0.14 ^b, xy^
	Ethanolic	23.61 ± 2.16 ^ac, xyz^	30.35 ± 0.56 ^ac, xyz^	88.99 ± 1.48 ^ac, xy^
	Aqueous	35.56 ± 1.43 ^b, z^	56.60 ± 1.18 ^ab, yz^	93.74 ± 0.09 ^b, xy^
*So*	Methanolic	39.41 ± 0.63 ^c, yz^	81.24 ± 1.02 ^bc, xyz^	94.90 ± 0.09 ^c, xy^
	Ethanolic	37.05 ± 0.81 ^c, z^	77.00 ± 1.31 ^ac, xyz^	95.21 ± 0.05 ^c, xy^
	Aqueous	31.64 ± 0.81 ^ab, xz^	53.09 ± 1.32 ^ab, z^	92.43 ± 0.14 ^ab, xy^
*Sm*	Methanolic	32.89 ± 0.61 ^c, z^	61.16 ± 0.86 ^xyz^	94.98 ± 0.05 ^xy^
	Ethanolic	30.63 ± 0.59 ^c, xz^	63.76 ± 0.62 ^xyz^	95.09 ± 0.11 ^xy^
	Aqueous	37.59 ± 3.51 ^ab, yz^	62.52 ± 1.54 ^xyz^	93.74 ± 0.25 ^xy^
BHA		36.20 ± 0.66 ^z^	55.43 ± 1.38 ^z^	75.43 ± 1.16 ^yz^
BHT		32.73 ± 0.60 ^z^	52.01 ± 0.53 ^z^	71.90 ± 1.86 ^xz^
Ascorbic acid		89.42 ± 0.17 ^xy^	too high ^xy^	too high ^xy^

The following coefficients were used to highlight the statistical difference (*p* < 0.05) between the extracts of the same plant species: ^a^—vs. methanolic extract, ^b^—vs. ethanolic extract, ^c^—vs. aqueous extract; and between the extracts and positive controls: ^x^—vs. BHA, ^y^—vs. BHT, ^z^—vs. ascorbic acid.

## Data Availability

Not applicable.

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
