# Peer review of "A Study of Phytochemistry, Genoprotective Activity, and Antitumor Effects of Extracts of the Selected Lamiaceae Species"

_plants, 2021, doi:10.3390/plants10112306_

Round 1

Reviewer 1 Report

The manuscript entitled "A study of phytochemistry, genoprotective activity and anti-tumor effects of extracts of the selected Lamiaceae species" is a well-structured and easy to follow, which contains information relevant to the literature. However, in order to be published, I would have some recommendations.
The purpose of the study must be specified in the abstract.
2. Results and Discussion
This section may be divided by subheadings. It should provide a concise and precise description of the experimental results, their interpretation, as well as the experimental conclusions that can be drawn. - this paragraph must be deleted
Table 1 lacks statistical analysis. Standard deviation is not sufficient for a statistical analysis.
Table 4 incomplete statistical analysis

Reviewer 2 Report

Comments and Suggestions for Authors

the authors presented “A study of phytochemistry, genoprotective activity and anti-2 tumor effects of extracts of the selected Lamiaceae species”. The main objective of this study was to evaluate the antigenotoxic and antitumor potential of lamiaceae extracts.

General comments:

Introduction:

Although the authors clearly and concisely describe the antioxidant, antitumor, anti-inflammatory and other beneficial properties of lamiaceae extracts, no mention of ROS-induced DNA damage and the importance of antioxidants in protecting and preserving DNA integrity has been given. Authors should describe this to better justify the purpose of their study.

Result and Discussion:

The authors should provide a more persuasive explanation of the antitumor mechanism of action, the hypothesis  that since tumor cells divide faster than normal ones, they are more sensitive to the effects of these extracts... should be supported by literature data.

Furthermore, the advantages and disadvantages of in vitro results compared to in vivo research should be described.

Specific comment

Result and Discussion:

Please include representative images of the comet assays and describe how % tail intensity was calculated. How were selected nuclei to include?

50 nuclei are not enough for an adequate DNA integrity evaluation in a comet assay, how was the experiment replicated? maybe you mean 50 nuclei per gel in duplicate or triplicate? please specify this.

Figure 2: consider replacing “induction rate” with “SOS induction rate”

Please add p-value in all figure legends.

Round 2

Reviewer 1 Report

The authores responded to all my comments